# Bipolar Patients and Bullous Pemphigoid after Risperidone Long-Acting Injectable: A Case Report and a Review of the Literature

**DOI:** 10.3390/brainsci11111386

**Published:** 2021-10-22

**Authors:** Michele Fabrazzo, Mariangela Boccardi, Salvatore Cipolla, Raffaele Galiero, Claudia Tucci, Francesco Perris, Ester Livia Di Caprio, Francesco Catapano, Ferdinando Carlo Sasso

**Affiliations:** 1Department of Psychiatry, University of Campania “Luigi Vanvitelli”, Largo Madonna delle Grazie 1, 80138 Naples, Italy; mariangela.boccardi@email.it (M.B.); salvatore2211@gmail.com (S.C.); claudiatucci3@gmail.com (C.T.); francesco.perris@unicampania.it (F.P.); esterlivia.dicaprio@unicampania.it (E.L.D.C.); francesco.catapano@unicampania.it (F.C.); 2Department of Advanced Medical and Surgical Sciences, University of Campania “Luigi Vanvitelli”, Piazza Luigi Miraglia 2, 80138 Naples, Italy; raffaele.galiero@unicampania.it (R.G.); ferdinandocarlo.sasso@unicampania.it (F.C.S.)

**Keywords:** bipolar disorder, risperidone LAI, antipsychotics, bullous pemphigoid, consultation–liaison psychiatry, neuroinflammation

## Abstract

Neuropsychiatric disorders are found to be associated with bullous pemphigoid (BP), an autoimmune subepidermal blistering disease. Antipsychotics have emerged as possible inducing factors of BP. However, large sample studies concerning BP associated with antipsychotics, as well as with specific mental disorders, are still lacking. Our review retrieved a few clinical studies and case reports on the topic, producing controversial results. We report for the first time a bipolar patient case presenting BP following five-month therapy with risperidone long-acting injectable (LAI). We hypothesize that the dermatological event is associated with the medication administered. The issue emerged during psychiatric consultation and was confirmed by histological examination, direct and indirect immunofluorescence studies, plus positive plasma and cutaneous BP180 and BP230 IgG. Neurodegeneration or neuroinflammation might represent a primary process leading to a cross-reactive immune response between neural and cutaneous antigens and contributing to self-tolerance failure. Furthermore, the time sequence of the shared biological mechanisms leading to clinical manifestations of the neuropsychiatric disorder and BP remains undefined. BP comorbid with bipolar disorder might occasionally represent a serious health risk and affect patients’ physical and psychosocial quality of life. Thus, clinicians treating psychiatric patients should consider BP as a possible adverse effect of psychotropic medications.

## 1. Introduction

Bipolar disorder (BD) is presently recognized as a multisystem condition affecting mood and cognitive, endocrine, autonomic, and sleep functions [1]. In addition, patients with BD may experience high rates of cardiovascular, pulmonary, infectious, and metabolic comorbidities, all associated with a decreased life expectancy and a worse quality of life (QoL) [2,3].

Recent evidence suggests that inflammation and immune dysregulation may play a remarkable role in the neuroprogression of BD [4]. The pathophysiological mechanisms underlying the disorder might be mediated by chronic systemic inflammation, as well as neuroinflammatory processes leading to the involvement of activated microglia, decreased neurogenesis, and increased apoptosis [5]. Patients with autoimmune diseases are at a greater risk for BD, and, conversely, patients with BD show a high incidence of autoimmune comorbidities [6]. Indeed, several autoimmune disorders are reported in association with BD, including Guillain-Barré syndrome, Crohn’s disease, autoimmune hepatitis, multiple sclerosis, rheumatoid arthritis, autoimmune thyroiditis, and systemic lupus erythematosus [7,8,9,10]. 

Bullous pemphigoid (BP) is an autoimmune chronic dermatologic disease characterized by a severe clinical presentation, relapses, and prolonged immunosuppressive treatment that rarely appears to be associated with BD. 

Previously, no systematic review has focused on the epidemiological, clinical, and pharmacological aspects of BP in comorbidity with BD. Instead, a number of studies exist on a significant association between BD and pemphigus, another autoimmune bullous disease with a pathogenesis similar to that of BP [11]. 

Most studies reported that the onset of the skin disorder usually followed the clinical neuropsychiatric manifestations [12,13], thus negatively affecting patients’ physical and psychosocial QoL [14,15,16,17]. On the other hand, Rania et al. [18] reported that medications used to treat patients with BP or bullous disorders might increase the risk of the onset of psychiatric disorders.

The present article aimed to analyze the potential immune–inflammatory link between BD and BP and identify the role of antipsychotics as a possible triggering factor of BP. To this aim, we illustrated the challenging case of a bipolar patient experiencing BP after the treatment with risperidone long-acting injectable (LAI).

## 2. Bullous Pemphigoid: Epidemiological, Pathogenetic, and Clinical Features

BP is a common autoimmune subepidermal blistering disease of the skin, defined by autoantibody formation against components of the hemidesmosome that collectively comprise the basement membrane zone [19]. In most cases, BP affects the patients’ skin, although eyes, mouth, and genitals may occasionally be involved [20]. 

The autoimmune mechanisms underlying BP now appear established. Clinical observations and experimental evidence have shown that patients possess autoantibodies and autoreactive T cells directed against two well-characterized self-antigens, primary components of the hemidesmosome adhesion complex that allow the basal cells of the epidermis to adhere to the basement membrane [21]: BP antigen 180 (BP180), also known as BPAG2 or type XVII collagen, and BP antigen 230 (BP230), further identified as an epithelial isoform of BPAG1 (BPAG1e). The upstream and downstream mechanisms related to the pathogenesis of BP remain a complex and partly defined area. So far, an imbalance between T regulatory cells (Tregs) and autoreactive T helper (Th) cells is considered as the main pathogenic factor triggering the autoimmune response in BP patients [22]. 

The contributory role of signaling pathways that foster the B cell stimulation remains controversial, such as the activation of Toll-like receptors (TLRs) and subordinate inflammatory mechanisms responsible for blister formation, such as Th17 axis stimulation and the activation of the coagulation cascade [23,24]. Similarly, the pathological mechanisms implied in the loss of dermal-epidermal adhesion secondary to autoantibodies binding are not thoroughly explained [22].

Almost all BP patients have circulating immunoglobulin (IgG) autoantibodies that bind specifically to the immune dominant non-collagenous NC16A extracellular domain of BP180. Both in vitro and in vivo studies provided strong evidence for the pathogenic role of BP180 autoantibodies [23]. Furthermore, it is reported that tissue injury occurs in the presence of antibody-antigen complexes and the serum levels of autoantibodies to BP180 NC16A correlate with the skin disorder severity [24]. During the acute flare of the disease, the bullous lesions appear stretched, diffused, and itchy. Afterward, the lesions evolve from erythema into urticarial plaques, and finally, subepidermal blisters erosions completely resolve. Thus, the inflammation resolution can be monitored in situ [25]. 

BP commonly proves to display a better prognosis compared to other chronic skin disorders [26]. Nevertheless, several studies of the last decade reported that BP patients, despite specific treatments, showed a prognosis similar to that of end-stage heart disease, with more than 40% dying within 12 months from the onset. Much of the mortality might be caused by age and general condition or be secondary to treatment with corticosteroids and other immunosuppressive agents [27,28].

BP is regarded a disease that mainly affects older individuals, with onset after 75 years of age, though the literature describes a few cases on young adults or children [29]. Most studies report a female prevalence, with a female-to-male ratio ranging from 1.0 to 5.0 [30,31], and an incidence rate higher in women up to 75 years. The incidence rate becomes higher in male patients >75 years [32].

Kridin and Ludwig [29] reported that the annual incidence range of BP is estimated to be 2.4–23 cases per million in the general population and 190–312 cases per million in patients ≥80 years. In addition, the authors observed an upward trend of BP incidence with a range of 1.9 to 4.3-fold during the last two decades.

## 3. Neuropsychiatric Disorders and Bullous Pemphigoid: A Potential Neuroimmunological Origin

BP is frequently concurrent with neuropsychiatric diseases, though such association remains controversial due to the limited number of large sample studies.

The neurologic disorders associated with BP proved to be dementia, Parkinson’s disease, cerebrovascular disorders, and epilepsy, with multiple sclerosis showing the highest risk [12].

A neurologic disorder affecting the patient appears to increase the risk for consequent BP, whereas the risk of developing a neurologic disease following BP is not equally clear [12,33,34]. BP might also be associated with psychiatric diseases such as schizophrenia, unipolar or bipolar disorders, schizotypal and delusional disorders, in addition to personality disorders. Nevertheless, the risk ratio is usually lower than neurologic diseases [12,35,36,37].

Forsti et al. [12] highlighted that psychiatric diagnoses are mainly formulated between 7 and 11 years before the onset of BP.

The epidemiological association between neurologic or psychiatric disorders and BP induced us to hypothesize that neurodegeneration and neuroinflammation are the primary processes leading to a cross-reactive immune response between neural and cutaneous antigens and contributing to self-tolerance failure. Indeed, both skin and nervous tissue derive from the ectoderm. The autoantigens BP180 and BP230 are expressed in the skin and highly present in the brain, specifically in the cortex and hippocampus [38]. However, Forsti et al. [12] reported that a few subpopulations of patients with Alzheimer’s or Parkinson’s disease possess circulating IgG autoantibodies against BP180 that do not bind to the cutaneous basement membrane or cause BP-like symptoms. Therefore, the production of primary neural autoantibodies might represent a predisposing factor for psychiatric or neurologic disorders that gradually lead to skin disease, considering that neurologic-psychiatric disorders generally precede BP onset [12].

Actually, most studies on BP associated with neurologic or psychiatric disorders reported that the onset of the skin disease usually followed the clinical psychopathological manifestations [12,13]. However, the time and cause relationship between the biological mechanisms underlying the association of the clinical onset of the neuropsychiatric disorders with BP remains unsettled to date. 

Beck et al. recently hypothesized that the time interval between the occurrence of the neurologic disease and the onset of BP might be due to the gradual exposure to the autoantigens (BP180 and BP230) as the neurologic disease progresses. Therefore, the risk occurs of developing symptoms in other tissues, such as the skin and mucous membranes containing the same antigens [39]. 

Furthermore, an issue to investigate more widely is whether exogenous triggering factors may cause self-tolerance failure and consequently develop dermatological and neuropsychiatric disorders. A hypothesis is formulated that in BP patients, a few predisposing genetic factors, such as class II HLA (e.g., HLA-DQβ1*0301) [40], and environmental factors (UV radiation, traumas, and drugs), might contribute to losing the immune tolerance toward such antigens of the dermal–epidermal junction [22]. Furthermore, it is suggested that the imbalance between autoreactive T helper (Th) and T regulatory cells (Treg), as well as a T cell-independent activation of toll-like receptor (TLR) system, may induce B cell stimulation, with consequent BP autoantibodies secretion [39,40,41]. In parallel, the Th17 pathway activation appears to maintain the inflammatory cascade caused by the humoral hyperactivation by triggering Th2 response, recruiting neutrophils and eosinophils, and stimulating proinflammatory cytokines and proteolytic enzymes release [39,42]. 

To date, only a few studies have examined the potential association between BP and psychiatric conditions [12,35]. However, a retrospective case-control study found that the highest risk existed for schizophrenia (OR = 2.7, 95% CI 2.0–3.5) and personality disorders (OR = 2.2, 95% CI 1.3–3.3) [12]. 

With regard to such matter, Bastuji-Garin et al. [36] conducted a multicenter case-control study that included 201 incident BP cases, among which only eight with a psychiatric diagnosis, namely unipolar or bipolar disorders. Differently, Teixeira et al. [43] reported the main association between BP and neurologic diseases in a cohort of 77 patients, with no relationship with depression or other affective disorders compared to matched controls without a history of BP. Nevertheless, based on the current literature, the association between BP and psychiatric disorders remains controversial [35].

## 4. Antipsychotics as Potential Inducers of BP

### 4.1. Drugs as Triggering Factors of BP

Triggering factors of BP are mainly physical agents, viral infections, and drugs identified in no more than 15% of genetically predisposed patients [44].

Drugs may act as triggers by either modifying the immune response or altering the antigenic properties of the epidermal basement membrane, thus provoking a sensitization during which are involved primary stimulation and expansion of drug-specific T lymphocytes. Such a process may affect the sole T cells or T and B cells with consequent formation of drug-specific antibodies (mostly IgE). Drugs are considered to act as haptens or prohaptens, being too small to elicit an immune response. After primary sensitization to a causative drug, a second exposure causes affected T cells and antibodies to enter the elicitation phase [45].

Coombs and Gell’s classification divides allergies into four pathophysiological types, namely the immediate (type I), cytotoxic (type II), immune complex-mediated (type III), and delayed (type IV) reactions [46]. Most drug allergies belong to type I or IV, being type II and III rarely observed. When the primary drug sensitization caused the formation of drug-specific IgE, the renewed contact with small amounts of antigens (drugs) might induce symptoms.

Overall, drug-induced BP is characterized by a younger age at the onset compared to the idiopathic form, and also by cutaneous eruptions persisting up to three months after discontinuing the culprit medication [47].

Lesions usually present as itchy and tense bullae on normal skin, more infrequently on erythematous/urticarial bases. In detail, the most affected areas are typically the trunk, the limbs (commonly the legs), and the face. The mucous membranes involvement, when occurring, generally appears mild. The erosions following the rupture of the bullae heal spontaneously without scarring.

The patients affected by BP occasionally show marked serum eosinophilia, in addition to IgG and/or C3 linear deposits, which are localized by direct immunofluorescence [32]. Actually, in most subepidermal autoimmune blistering conditions, autoantibodies form deposits that cause the release of proteolytic enzymes by activating the complement cascade, which destroys the basement membrane [32].

Furthermore, BP appears to be associated with the use of oral and topical medications, though no specific antigens are identified, being presumably the same as in the spontaneous form. The clinical, histopathological, and immunological features of the idiopathic and drug-induced forms of BP are similar. Consequently, the differential diagnosis comparing the two forms may be complex. The diagnosis of a drug-induced BP is to be formulated carefully in elderly patients whose daily pharmacotherapy may vary over time.

So far, two types of drug-induced BP have been acknowledged. The first is an acute and self-limited condition that regresses when the drug is withdrawn and is defined as “drug-induced BP proper”. The second is a chronic, persistent and severe condition evolving as the classic BP, and is defined as “drug-triggered BP” [47]. In addition, in a subject already sensitized to a given agent, cross-reactivity may develop toward another agent of the same class or having a similar chemical structure. Accordingly, the immunological response to target the skin and provoke bullae formation might be caused by one of the following factors: the drug-induced anti-basement membrane antibodies formation, the dysregulation of the immune system, or the molecular mimicry [26].

### 4.2. Antipsychotics and Bullous Pemphigoid

Antipsychotics proved to be comprised of several classes of drugs associated with the onset of BP [48,49,50]. For example, Bastuji-Garin et al. [34] mentioned the chronic use of spironolactone (OR, 2.30; 95% CI, 1.20–4.46) or aliphatic phenothiazines (OR, 3.70; 95% CI, 1.21–11.34) as the most potential cause of BP. Indeed, clinically relevant cutaneous reactions may develop following the use of different classes of psychotropic agents, with a prevalence rate of approximately 5% due to typical and atypical antipsychotics (Table 1) [50,51].

Psychotropic medications may contribute to adverse cutaneous manifestations with a spectrum of varying severity degrees, ranging from benign to rare, life-threatening skin reactions such as toxic epidermal necrolysis and Stevens-Johnson syndrome [50,51]. Such a class of drugs may also aggravate pre-existing skin conditions or trigger the manifestations of a latent state, as for example, lithium-induced psoriasis [52]. Cross-sensitivity reactions may pertain to medications of the same class or of different classes. Therefore, it seems essential to accurately monitor the administration of the psychotropic drug to early detect and manage the adverse cutaneous reactions, especially during the initial months of treatment [52,53].

Chlorpromazine is reported to be the first generation antipsychotic that primarily causes hyperpigmentation (1–2.9%) and photosensitivity (up to 25%) (Table 1) [52]. More frequently, grey pigmentation changes appear in women who have taken the drug continuously for more than three years. When chlorpromazine is exposed to UV light, it transforms into a free radical form, which combines with melanin, and is trapped in the melanocytes. Furthermore, chlorpromazine can function as an activator of tyrosinase, an enzyme used to produce melanin. Also, risperidone cutaneous reactions are thought to be mediated by a similar mechanism [54].

Most adverse cutaneous reactions appear mild and may be easily treated without exposing patients to a serious health risk. However, such reactions may considerably impact treatment compliance.

The most frequently cutaneous adverse effects of antipsychotic medications include exanthematous eruptions, skin pigmentation changes, photosensitivity, urticaria, and pruritus (Table 1). Therefore, the withdrawal of the causative medication and the switch to a different class of agents are decisive steps to minimize morbidity and promptly identify severe drug reactions.

Atypical antipsychotics such as clozapine, olanzapine, quetiapine, risperidone, and paliperidone, similarly to the typical antipsychotics, are documented to cause skin reactions (Table 1). Nevertheless, cutaneous adverse drug eruptions are less frequent when administering second-generation antipsychotics (SGAs) [52]. Adjunctive therapy with atypical antipsychotics is widely used to treat acute bipolar depressive or manic episodes to foster treatment adherence and control psychotic symptoms typical of the manic phases [53]. In addition, atypical antipsychotics may also inhibit the monoaminergic reuptake transporter, contributing to stabilizing further the mood episodes [55,56,57]. The current therapy of BD also includes new-generation antipsychotics (NGAs), which may increase the probability of adverse skin drug reactions.

Erythema multiforme (EM) is an acute reactive mucocutaneous disease of the skin and mucous membranes associated with antipsychotics, particularly chlorpromazine and other traditional neuroleptics. Further case reports also describe EM, as associated with ziprasidone and risperidone [58,59], differently from paliperidone, reported to cause toxic epidermal necrolysis and exanthematous rash [60,61]. In addition, a case of diffuse erythematous and maculopapular skin rash is also reported due to the administration of risperidone LAI in a young male patient, previously treated with oral risperidone with no skin side effects [62].

Olanzapine is described to be associated with severe generalized pruritic skin eruptions as a manifestation of hypersensitivity syndrome [63] and to cause leukocytoclastic vasculitis displaying as erythematous skin eruptions [64].

Another rare adverse cutaneous reaction associated mainly with risperidone, clozapine, ziprasidone, droperidol, and chlorpromazine is angioedema, which is characterized by the edema of the deep dermal and subcutaneous tissues [65].

The literature describes rare cases of aripiprazole-induced skin reactions. For example, Nath et al. [66] illustrated a case of a young male patient affected by schizophrenia, who developed a skin rash after starting aripiprazole (20 mg/day) that remitted after drug withdrawal.

In Table 2, we summarized all the studies retrieved in a thorough Pubmed search until 6 August 2021, using the keywords “bullous pemphigoid” and “antipsychotics”, concerning the potential association between BP and antipsychotics. In all the studies, enrolled patients mainly were females, with a mean age ≥ 74 years. Psychiatric diagnoses and antipsychotic treatments were not always specified.

Bastuji-Garin et al. [67] selected only 16 patients over a total of 116 incident cases of BP presenting clinical manifestations of the skin disorder after antipsychotic treatments (Table 2). Bastuji-Garin et al. [36], in another multicenter case-control study, identified retrospectively, 201 incident cases of BP, among which only 24 were treated with non-specified antipsychotics and 13 with phenothiazines. The antipsychotic-treated patients had a diagnosis of unipolar or bipolar disorders.

Wijeratne and Webster [68], on the other hand, illustrated a clinical BP case after exposure to risperidone in an old male patient with dementia and psychotic symptoms. Instead, an additional clinical report described the case of a BP female patient after treatment with thioridazine hydrochloride and flupentixol decanoate [69]. Finally, Varpuluoma et al. [70] reported data from a retrospective study analyzing 3397 cases of BP, with a small percentage of patients treated with a different type of antipsychotic (range 0.4–5.1%), being the risperidone in the highest percentage (Table 2). In contrast, Lloyd-Lavery et al. [71], in a case-control study including 86 cases of BP, reported no significant differences in patients using antipsychotics compared with controls.

**Table 2 brainsci-11-01386-t002:** Clinical studies and cases diagnosing bullous pemphigoid (BP) in association with antipsychotic treatments.

Source	Study Design, Numbers of Participants	Gender (%), Mean Age Years (±sd)	Psychiatric Diagnoses	Number of Patients (%) and Antipsychotics Treatments
**Bastuji-Garin et al., 1996 [67]**	Multicenter prospective case-control study, 116 incident cases of BP	F (50%), M (50%) 79.2 (10.1)	Not specified	18 (15.5%) treated with NSA
**Wijeratne and Webster, 1996 [68]**	Case report 1 BP patient	M 74 years	Dementia with psychotic symptoms	Risperidone (4 mg/day)
**Mehravaran et al., 1999 [69]**	Case report 1 BP patient	F 73 years	Not specified	Thioridazine, hydrochloride and flupentixol decanoate
**Bastuji-Garin et al., 2011 [36]**	Multicenter case-control study, 201 incident cases of BP	F (65%), M (35%) 84.2 (8.7)	Unipolar and bipolar disorders	24 (11.9%) treated with NSA 13 (6.5%) treated with phenothiazines with aliphatic side chains
**Lloyd-Lavery et al., 2013 [71]**	Case-control study 86 cases of BP	F (59.3%), M (40.7%) 81.5 (9.7)	Not specified	6 (7.0%) treated with NSA No significant differences for antipsychotics
**Varpuluoma et al., 2019 [70]**	Retrospective study (Finnish Care Register for Health Care database, 1987–2013) 3397 cases of BP	F (59.7%), M (40.3%) 76.6	Not specified	34 (1.0%) Perphenazine 49 (1.4%) Haloperidol 97 (2.9%) Quetiapine 13 (0.4%) Sulpiride 172 (5.1%) Risperidone

F = female; M = male; NSA = not specified antipsychotics.

## 5. The Challenging Case of a Bipolar Patient Presenting Bullous Pemphigoid after Risperidone LAI

Consultation–liaison psychiatry is a recent subspecialty of psychiatry that concerns diagnosis, treatment, and prevention of psychiatric morbidity of patients affected by physical diseases. Correspondingly, managing a psychiatric patient who requires hospitalization for medical problems becomes troublesome for the physicians in charge due to their limited familiarity with psychiatric disorders and related pharmacotherapies. In addition, patients with mental health problems are often stigmatized and suffer a negative attitude towards themselves [72].

In our paper, we describe the problematic case of a patient with a history of BD who experienced severe skin complications due to psychopharmacological treatment.

Our patient is a 71-year-old man with a psychiatric history of bipolar affective disorder initiated at the age of 37, after the end of a significant relationship. Moreover, a family psychiatric history was recorded in the patient’s anamnesis. He experienced a manic episode characterized by elevated mood, inflated self-esteem, decreased need for sleep, increased energy, and increased goal-directed activity, in addition to grandiose and persecutory delusions and behavioral disorganization. The patient was therefore hospitalized and treated with psychoactive drugs, which he did not specify. The following years were characterized by a partial psychopathological compensation associated with oscillating functionality. In addition, the patient reported several depressive episodes not treated with antidepressants. In 2016, a dose of 20 mg/day of olanzapine was introduced in the daily drug regimen of the patient, who acquired psychopathological and functional compensation. In July 2019, he experienced a new severe manic episode characterized by excessive shopping and behavioral disorganization, in addition to grandiosity, persecutory, and poisoning delusions. Such symptoms affected the compliance of the patient, who was therefore hospitalized after a few weeks from the mental manifestations. In the daily therapy, risperidone was introduced at a dose of 4 mg/day. The acute episode terminated in September 2019, when the patient started a maintenance therapy with lithium carbonate (600 mg/day) and, from October 2019, with risperidone LAI 37.5/2 mL administered every two weeks. In January 2020, the patient presented psoriasiform lesions, presumably associated with lithium carbonate use; consequent to such an event, our patient was hospitalized. In a first psychiatric consultation, physicians prescribed removing the mood stabilizer and switching to valproate. About 20 days after initiating the medication, a second switch (to oxcarbazepine) became necessary due to worsening dermatologic lesions.

Simultaneously the patient initiated treatment with prednisone (37.5 mg/day) and azathioprine (50 mg/day), which were added to the psychotropic drugs already prescribed (risperidone LAI 37.5 mg/two weeks and oxcarbazepine 600 mg/day).

In February 2020, the patient was transferred from the psychiatry to the internal medicine unit, where BP was diagnosed after histological examination evidencing subepidermal cleft with eosinophils. In addition, direct immunofluorescence showed linear deposition of antibodies and complement (IgG and C3), indirect immunofluorescence detected circulating IgG antibodies against basement membrane proteins, and enzyme-linked immunosorbent assay quantified positive BP180 or BP230 IgG antibodies. Figure 1 shows urticaria-like red skin lesions of our patient without blisters (A) and post-blistering erosions localized to the lower left limb (B).

In March 2020, while staying home, the skin lesions extended to the whole body surface, and a bacterial infection appeared due to *Streptococcus pyogenes*. Consequently, the patient was conducted first to the emergency unit and then to the internal medicine department. During hospitalization in the emergency unit, a chest computed tomography (CT) and abdominal ultrasound (US) excluded paraneoplastic syndrome. Furthermore, blood tests revealed leucopenia with associated lymphocytosis and increased mononuclear cells, hyponatremia (129 mEq/L), decreased estimated glomerular filtration rate (eGRF) with increased plasma urea (99 mg/dL), and a rise in C-reactive protein (CRP) (42.75 mg/dL).

In March 2020, our patient was transferred to the internal medicine department of the University of Campania “Luigi Vanvitelli” to manage the complexity of the clinical case more comprehensively. Consequently, the patient developed autoimmune hemolytic anemia triggered by azathioprine that was promptly treated with blood transfusion.

The state of consciousness rapidly deteriorated until the patient slipped into a coma (Glasgow Coma Scale scores 2-2-4) and presented hypothermia combined with severe hypotension. The serious condition urged a transfer to the intensive care unit. Moreover, the risk of septic shock and pulmonary thromboembolism gradually arose and required therapy with broad-spectrum antibiotics combined with heparin. However, heparin caused autoimmune thrombocytopenia and was promptly discontinued to be replaced by rivaroxaban (20 mg/day).

Once the physical condition was recovered, the patient was retransferred to the internal medicine unit, where he showed a marked state of agitation, necessitating an additional psychiatric evaluation. The consultant psychiatrist prescribed haloperidol (2 mg/day), delorazepam (1.5 mg/day), and triazolam (0.250 mg/day). A hypothesis emerged suggesting a correlation between skin reactions and the depot antipsychotic injections, examined using the Naranjo Adverse Drug Reaction Probability Scale (NADRPS) [73]. The assessment score indicated a “highly probable” relationship between the adverse effects and risperidone LAI, which was consequently withdrawn.

Once the skin completely recovered, the patient was transferred to our psychiatric unit and administered a higher daily dose of 4 mg haloperidol, and the overall therapy remained unchanged. Approximately one month later, he was dismissed from our department and assigned to a long-term facility. During the follow-up, the psychopathological symptoms as an outpatient remained unvaried. Accordingly, the ongoing medication therapy was continued. After six months, the general conditions improved, and the patient reported feeling emotions and participating more extensively in all the facility’s daily activities. Moreover, he showed critical thinking about his delusional beliefs. The mood remained stable with no further oscillations.

## 6. Discussion

The present article aimed to analyze a potential immune–inflammatory link between BD and BP and highlighted the role of antipsychotics as a possible triggering factor of BP. To this aim, we illustrated the challenging case of a bipolar patient presenting BP after a five-month therapy with risperidone LAI.

The case we described represents a comorbid condition exacerbated by severe sequelae resulting from administering a specific atypical long-acting antipsychotic that presumably acted as a triggering factor. Nevertheless, six months before BP clinical manifestation, our patient had taken oral solution risperidone with no adverse skin events. Therefore, we proposed that the LAI formulation of risperidone triggered the skin side effects, as also suggested by the result of the NADRPS [73]. It should be stressed that risperidone LAI is composed of microspheres of polylatide-co-glycolide (PLGA) polymers that are biodegradable, biocompatible, and nontoxic, with a history of long-lasting use. Such microspheres measure between 25 and 150 μm, a more significant size than the standard drug molecules, which might act as haptens or prohaptens that are too small to elicit an immune response [74]. In conclusion, a possibility exists that risperidone microspheres in our bipolar patient elicited an immune reaction.

Only a few clinical studies and case reports have indicated BP as a side effect associated with the use of first- or second-generation antipsychotics in patients with BD [35,36,37,67,68,69,70].

Instead, no reports of adverse cutaneous reactions associated with risperidone LAI are reported in psychiatric patients, particularly in those affected by BD. A report by Sidhu et al. [62] indicated risperidone LAI treatment associated with a diffuse erythematous and maculopapular skin rash in a young male patient whose psychiatric diagnosis was not specified. Both the patient described in the report and our bipolar patient did not show skin side effects when initially treated with oral risperidone. Differently, dermatological side effects appeared when both patients were subsequently treated with risperidone LAI. Furthermore, our finding of an association between BP and risperidone is in line with another case reported by Wijeratne and Webster [68], which described the onset of BP after oral risperidone administration in a patient diagnosed with dementia and treated with such medication prevalently for psychotic symptoms. Likewise, Varpuluoma et al. [70] recently reported that oral risperidone was associated with BP in neurologic-psychiatric patients whose diagnosis was not specified.

Bastuji-Garin et al. [36] reported that a rise in BP incidence has been recently documented in Europe. Among the risk factors of BP, the authors indicated neurologic disorders, particularly degenerative diseases that may involve autoimmune mechanisms such as Parkinson’s and Alzheimer’s diseases, psychiatric disorders (unipolar and bipolar disorders), bedridden conditions, and chronic use of several drugs. The authors also suggested that immobility, muscle weakness, or low self-sufficiency, also evidenced in our bipolar patient, may have a role in the pathogenesis of BP lesions. Such findings comply with our hypothesis about neurodegenerative or neuroinflammatory mechanisms operating in neuropsychiatric disorders. Moreover, such mechanisms might predispose some patients to BP after introducing triggering factors such as drugs (i.e., antipsychotics). In Table 3, we summarized the conditions frequently considered risk factors in neuropsychiatric patients.

The increasing use of atypical antipsychotics as components of the mood-stabilizing pharmacotherapies for treating bipolar patients might probably predispose patients to BP clinical manifestations more frequently. Risk factors may have considerable implications in managing BP patients. Therefore, further studies on a larger population are needed to explore more widely the issue, especially the time and cause relationship of the comorbid processes.

It remains unclear whether BP might act either as a prodrome of the neurologic-psychiatric disorders that subsequently activate neuroinflammation, or as a primary CNS condition extending to peripheral organs and leading to a cross-reactivity state.

In this regard, Forsti et al. [12,35] concluded that the association between BP and a few psychiatric disorders, particularly schizotypal and delusional, was bidirectional. Nevertheless, the risk of BP onset is more frequently consequent to a psychiatric disorder than vice versa.

Various studies of the last 20 years showed that the increased inflammation and hyperactivity of the hypothalamic–pituitary–adrenal (HPA) axis and sympathetic nervous system (SNS) are the biological findings most frequently shared by skin chronic inflammatory diseases and mood disorders [75,76,77]. In additional studies, other shared contributors to both conditions emerged: the disrupted secretion of the melatonin’s sleep hormone [78] and the decreased plasma concentration of vitamin D_3_ [79].

Evidence supports that inflammation, immune dysregulation, and brain involvement may share biological origins, specifically, overactivation of innate immunity leading to a cytokine-mediated inflammatory response and through a potential dysregulation of the HPA axis [80]. In addition, the biological link between physical and mental diseases is given by the enduring “cytokines’ storm”, which may activate brain microglia, proinflammatory mediators, blood–brain barrier permeability, and leukocyte extravasation, thus contributing to neuroinflammation [81,82].

As occurs for other skin inflammatory diseases, cytokines may be involved in the inflammatory process underlying BP, though the role of the key regulator molecules remains only partially explicated. For example, in a recent study [83], interleukin (IL)-17A emerged as having a central function.

An additional element that contributes to forming blisters may be associated with a member of the tumor necrosis factor (TNF) superfamily, the so-called TNF-like weak inducer of apoptosis (TWEAK), which stimulates the fibroblast growth factor-inducible-14 (Fn14), an interacting receptor of the TNF receptor superfamily (TNFRSF). Indeed, Liu et al. [84] showed that TWEAK serum levels inversely correlated to BP180 expression and cellular adherence. Moreover, TWEAK activation proved to trigger inflammation via extracellular signal-regulated kinases (ERK) and nuclear factor kappa-light chain-enhancer of activated B cells (NF-kB) pathways.

Most of the studies we analyzed showed a few limitations in detecting the interrelationships between the psychiatric diagnoses and the dermatologic symptoms of BP. The factor that makes it troublesome to manage the patient is the concurrence of diseases belonging to various medical branches, such as psychiatry, dermatology, and internal medicine. In addition, apart from psychiatrists and dermatologists, the physicians of other specializations show limited familiarity with the related diagnosis, assessment tools, and medication treatments. Furthermore, most studies on skin side effects after antipsychotics include small cohorts of patients, mainly aged ≥ 75 years, and only a few young patients.

The combined use of oral mood stabilizers and other antipsychotics might appear as confounding factors in our case. Nevertheless, it appears evident that BP worsened the 12 weeks following the last injection of risperidone LAI, whose withdrawal resulted in improving our patient. The exacerbated state of the patient finally required a different type of antipsychotic.

More extensive prospective studies on psychiatric patients are needed to investigate the association of BP with neuroinflammation and the molecular mechanisms leading to derange the immunological tolerance against BP autoantigens in the human brain.

Inflammation and immune dysregulation act significantly in BD neuroprogression. Therefore, further studies should include patients with mood disorders to explore the timing and function of autoimmunization against neural BP autoantigens in the pathogenesis of both disorders.

## 7. Conclusions

BD and BP are medical conditions associated with high functional impairment and distress, also characterized by high rates of health complications, contributing to a decreased life expectancy and a worse QoL [2,14,85]. Therefore, early identification of severe cases and complications of BP might decrease the morbidity and mortality caused by the disease. That is why we considered it essential to investigate and summarize the literature on the complications of BP in patients after antipsychotic treatment, also describing the case of a bipolar patient experiencing BP after risperidone LAI. Indeed, we highlighted the potential role of inflammatory components in both BD and BP, and of antipsychotics as inducing factors of skin side effects. Furthermore, we revealed that bipolar patients using antipsychotics for mood and psychotic symptoms are at greater risk of developing severe dermatologic complications. Dermatologists and psychiatrists should accordingly cooperate to detect the onset of BP in mental health patients to prevent comorbidity exacerbation.

## Figures and Tables

**Figure 1 brainsci-11-01386-f001:**
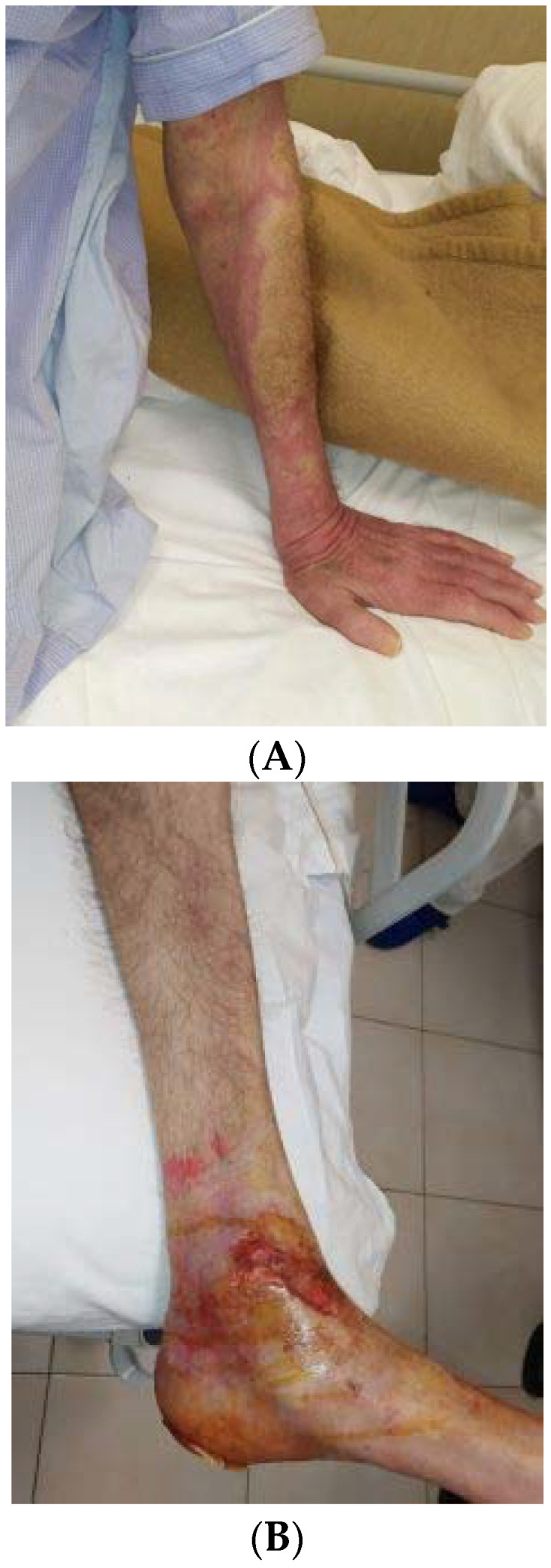
Bullous pemphigoid lesions present in our bipolar patient as urticaria-like red skin without blisters (**A**). Post-blistering erosions localized to the lower left limb (**B**).

**Table 1 brainsci-11-01386-t001:** Cutaneous drug reactions (CDRs) associated with the use of first-generation antipsychotics (FGAs), known as typical, and second-generation antipsychotics (SGAs), known as atypical.

	Antipsychotics	Dermatologic Features of CDRs
**FGAs** **(typical)**	Chlorpromazine	Pruritus, drug-induced pigmentation, exanthematous reactions, urticaria, exfoliative dermatitis, photosensitivity, erythema multiforme, Stevens-Johnson syndrome, angioedema
	Fluphenazine	Pruritus, exanthematous reactions, urticaria drug-induced pigmentation, exanthematous reactions
	Flupentixol	Drug-induced pigmentation, exanthematous reactions, bullous pemphigoid
	Haloperidol	Exanthematous reactions, fixed drug eruptions, acneiform eruptions, drug-induced pigmentation, alopecia, exanthematous reactions, photosensitivity, hyperhidrosis
	Perphenazine	Pruritus, exanthematous reactions, urticaria, drug-induced pigmentation
**SGAs (atypical)**	Aripiprazole	Exanthematous reactions, alopecia, acneiform eruptions, onycholysis
	Clozapine	Pruritus, exanthematous reactions, urticaria, drug-hypersensitivity vasculitis, drug-induced pigmentation, erythema multiforme, photosensitivity, Stevens-Johnson syndrome, angioedema
	Olanzapine	Pruritus, exanthematous reactions, urticaria, fixed drug eruptions, photosensitivity, drug-induced pigmentation, alopecia, seborrheic dermatitis, hyperhidrosis, drug-hypersensitivity vasculitis, exanthematous reactions
	Quetiapine	Pruritus, exanthematous reactions, fixed drug eruptions, photosensitivity, acneiform eruptions, drug-induced pigmentation, seborrheic dermatitis, hyperhidrosis, exfoliative dermatitis, psoriasiform reactions, angioedema
	Ziprasidone	Exanthematous reactions, exfoliative dermatitis, urticaria, photosensitivity, alopecia, angioedema
	Paliperidone	Toxic epidermal necrolysis, exanthematous rash
	Risperidone	Pruritus, exanthematous reactions, urticaria, alopecia, drug-induced pigmentation, acneiform eruptions, seborrheic dermatitis, hyperhidrosis, erythema multiforme, exfoliative dermatitis, psoriasiform reactions, photosensitivity, angioedema, bullous pemphigoid

**Table 3 brainsci-11-01386-t003:** Bullous pemphigoid risks factors in patients with neuropsychiatric disorders.

**Elderly age:** usual onset after 75 years of age
**Pre-existing skin diseases:** psoriasis, atopic dermatitis, and lichen
**Bedridden conditions:** immobility, muscle weakness, low self-sufficiency
**Severe cognitive impairment**
**Polypharmacotherapy:** loop diuretics, spironolactone, dipeptidyl peptidase-IV inhibitors, and antipsychotics
**Pharmacodynamic/pharmacokinetic drug interactions**

## Data Availability

Data presented in this review are available in the tables; clinical data of the included case-report are available at the Department of Advanced Medical and Surgical Sciences and the Department of Psychiatry of the University of Campania “Luigi Vanvitelli, Naples”.

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
