# Peer review of "Bipolar Patients and Bullous Pemphigoid after Risperidone Long-Acting Injectable: A Case Report and a Review of the Literature"

_brainsci, 2021, doi:10.3390/brainsci11111386_

Round 1

Reviewer 1 Report

This case report lacks important details, has a paucity of learning points and the writing also requires a close edit for language and grammar.

Specific comments:

  1. As per journal guidelines for authors, the abstract should be a total of about 200 words maximum.
  2. Please change "Neurologic-psychiatric disorders" to "Neuropsychiatric disorders".
  3. "Previously, no systematic revision" - do you mean "systematic review"?
  4. "Most studies reported that the onset of the skin disorder usually followed the clinical neurologic-psychiatric manifestations" - this is not true, in the study by Rania et al. (2020), although the temporal patterns indicate bidirectionality, the risk of a psychiatric disorder preceding bullous pemphigoid (BP) or all bullous disorders (ABD) was greatest.
  5. "neurlogic" - spelling error.
  6. "Inducing factors of BP are mainly physical agents, viral infections, and drugs identified in no more than 15% of genetically predisposed patients [44]" - Bullous pemphigoid (BP) is the most frequent autoimmune subepidermal blistering disease provoked by autoantibodies directed against two hemidesmosomal proteins: BP180 and BP230. Its pathogenesis depends on the interaction between predisposing factors, such as human leukocyte antigen (HLA) genes, comorbidities, aging, and trigger factors. Several trigger factors, such as drugs, thermal or electrical burns, surgical procedures, trauma, ultraviolet irradiation, radiotherapy, chemical preparations, transplants, and infections may induce or exacerbate BP disease. Rather than say inducing, triggering or predisposing factors would be more accurate.
  7. "The patient had been treated with oral risperidone with no skin side effects, as occurred in our bipolar patient" - please rephrase this.
  8. An inflammatory component, which plays a greater role in BP than in other autoimmune blistering disorders. It is driven by the action of polymorphonuclear cells. Several lines of evidence suggest the involvement of microglial activation and increased systemic inflammation in the pathogenesis of bipolar disorder (citation: ncbi.nlm.nih.gov/pmc/articles/PMC6895819).
  9. Please change "Therefore, we tend to suppose that" to "Therefore, we proposed that".
  10. What is the take-home message of this case report? These should be more succinctly presented.

Reviewer 2 Report

the authors have written a very interesting clinical case that explains the possible association between a rare autoimmune disease (bullous pemphigoid) and the use of atypical antipsychotics in a patient with bipolar disorder. The case report is well written and proposes an interesting etiological hypothesis for both pathologies. We thought that the article can be useful for psychiatrist and neurologist that prescribed antipsychotics frequently.

We have only a few comments for the authors.

We recommend to include a figure or a picture about the skin injuries produces for the BP, the readers of the journal normally are not dermatologist so it can be good to know when they can suspect this illness.

In the case report; it is not clear what happen with mood stabilizers (first lithium, second valproate and third oxcarbazepine) is the patient receiving now a mood stabilizer? Is he taking haloperidol for a long time? Haloperidol produce extrapyramidal symptoms in several patients: is the patient sensitive to this side effects?

We recommend to improve the reading of the article including a summary table with the principal risk factors of patients with psychiatric disorders than can suffer a BP after the use of antipsychotics.

Round 2

Reviewer 1 Report

Thank you for the revisions.